# Med-K2N: Flexible K-to-N Modality Translation for Medical Image Synthesis

## Abstract

Cross-modal medical image synthesis research focuses on reconstructing missing imaging modalities from available ones to support clinical diagnosis. Driven by clinical necessities for flexible modality reconstruction, we explore K→N medical generation, where three critical challenges emerge: ①How can we model the heterogeneous contributions of different modalities to various target tasks? ② How can we ensure fusion quality control to prevent degradation from noisy information? ③How can we maintain modality identity consistency in multi-output generation? Driven by these clinical necessities, and drawing inspiration from SAM2's sequential frame paradigm and clinicians' progressive workflow of incrementally adding and selectively integrating multi-modal information, we treat multi-modal medical data as sequential frames with quality-driven selection mechanisms. Our key idea is to **"learn"** adaptive weights for each modality-task pair and **"memorize"** beneficial fusion patterns through progressive enhancement. To achieve this, we design three collaborative modules: **PreWeightNet** for global contribution assessment, **ThresholdNet** for adaptive filtering, and **EffiWeightNet** for effective weight computation. Meanwhile, to maintain modality identity consistency, we propose the Causal Modality Identity Module (**CMIM**) that establishes causal constraints between generated images and target modality descriptions using vision-language modeling. Extensive experimental results demonstrate that our proposed Med-K2N outperforms state-of-the-art methods by significant margins on multiple benchmarks. Source code is available at https://anonymous.4open.science/r/Med-K2N-74E7/.

## 1 Introduction

Medical imaging diagnosis requires integrating multiple modalities for accurate assessments Pichler et al. (2008); Adam et al. (2014). However, obtaining complete multi-modal data is challenging due to equipment availability, examination time, and patient constraints Thukral (2015); Staartjes et al. (2021). Cross-modal generation techniques are thus important for reconstructing missing modalities Shen et al. (2020); Roy et al. (2010); Sharma & Hamarneh (2019); Roy et al. (2016); Bowles & et al. (2016). Existing one-to-one translation methods struggle with diverse clinical scenarios, while flexible K→N mapping approaches better address real-world multi-modal synthesis requirements.

Different input modalities contribute differently to specific target synthesis. For instance, DWI shows strong correlation with T2 signals, while such dependence is weaker in CT synthesis Chartsias et al. (2018). Current approaches use uniform fusion strategies without assessing modality-specific value Havaei et al. (2016); Varsavsky et al. (2018); Goodfellow et al. (2016); Hinton et al. (2006); Ye et al. (2013); Vemulapalli et al. (2015); Goodfellow & et al. (2020). Consequently, discriminative features are diluted by noise, compromising performance in multi-task scenarios.

Recently, vision foundation models (e.g., SAM series Kirillov et al. (2023); Chen et al. (2023); Xiao et al. (2024)) show promise in multimodal processing by modeling data as sequential frames. Recent advances in segment anything models have demonstrated remarkable capa-

bilities in medical image analysis Hu et al. (2024); Huo et al. (2024); Li et al. (2024a;b); Jin et al. (2021); Kachole et al. (2023); Liao et al. (2025); Li et al. (2023a). However, existing methods focus on single-output scenarios, failing to meet clinical multi-output requirements. When extending these approaches to parallel multi-output generation settings, three core challenges emerge: *(1) Inadequate Modeling of Modality-Task Heterogeneity:* Current methods use uniform weighting without characterizing differential modality contributions to different tasks. *(2) Lack of Quality Control Mechanisms in Fusion Process:* Fusion strategies lack real-time evaluation of information integration effectiveness, potentially introducing degrading information. *(3) Absence of Modality Identity Consistency in Multi-Head Generators:* Multi-head generators produce incorrect modality features (e.g., T2-like features when T1 is expected).

To address these challenges, we propose **Med-K2N**, a quality-aware progressive fusion framework. Medical image synthesis has evolved from traditional statistical methods to deep learning approaches, with recent works exploring multimodal fusion strategies Havaei et al. (2016); Varsavsky et al. (2018); Chartsias et al. (2018); Dar et al. (2019); Liu et al. (2023). For challenge **(1)**, we design Three collaborative modules: **PreWeightNet** learns global contribution weights (determining whether to use a modality), **ThresholdNet** learns adaptive filtering thresholds (determining acceptance criteria), and **EffiWeightNet** for learning effective weights (determining where and at what intensity to perform fusion), with progressive fusion following "primary frame + auxiliary enhancement" strategy. For challenge **(2)**, we employ quality-driven adaptive decision mechanism, where each auxiliary modality is accepted only if it can improve generation quality, ensuring effective control of the fusion process through real-time quality assessment. For challenge **(3)**, we introduce a causality-based module, the Causal Modality Identity Module (**CMIM**), which leverages the causal relationship **"modality type → visual features → semantic expression."** By employing a medical domain pre-trained vision-language model, it establishes causal consistency constraints between generated images and target modality descriptive texts, avoiding modality identity confusion.

Experimental results demonstrate Med-K2N's superior performance across various K→N configurations with consistent improvements in objective metrics including PSNR and SSIM. Main contributions include:

- **Progressive multimodal fusion architecture** overcoming information dilution through stepwise enhancement strategy;

- **Collaborative multi-module generation framework with adaptive fusion mechanisms**. We construct a complete quality-driven closed loop from MultiScaleNet to TaskHeadNet, ensuring that only beneficial information is retained while designing CMIM to prevent modality identity confusion in generated outputs.

- **Flexible medical multimodal generation system** supporting arbitrary modality numbers(K2N), providing a practical and scalable solution for clinical multimodal image synthesis.

- **Comprehensive experimental validation** on Combined Brain Tumor and ISLES 2022 datasets, where Med-K2N achieves superior performance compared to most state-of-the-art methods.

## 2 METHOD

This paper aims to establish a medical cross-modal generation framework supporting arbitrary K→N mappings and proposes the Med-K2N architecture (Fig 1). Medical image synthesis has been extensively studied using various generative approaches Shen et al. (2020); Roy et al. (2010), with recent advances in deep learning Goodfellow et al. (2016); Hinton et al. (2006) enabling more sophisticated cross-modal translations Goodfellow & et al. (2020). Previous works have explored patch-based methods Roy et al. (2016), dictionary learning approaches Huang et al. (2016); Iglesias et al. (2013), and statistical methods Ye et al. (2013); Roy et al. (2013); Pan et al. (2018); Staartjes et al. (2021); Bowles & et al. (2016). Med-K2N consists of a LoRA-fine-tuned SAM2 image encoder, a progressive cross-modal

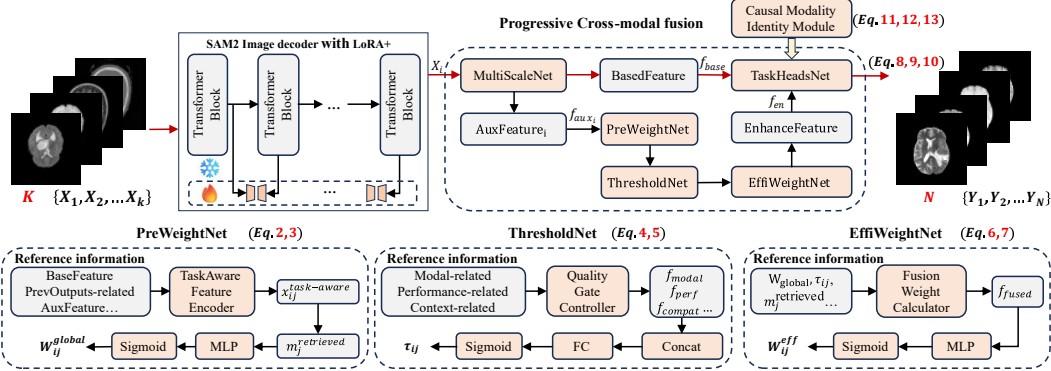

Figure 1: The overall framework of the proposed Med-K2N, it achieves flexible **K-to-N** modality mapping through a progressive fusion strategy of **"key frame baseline + auxiliary modality step-by-step enhancement"**, addressing modality-differentiated modeling and identity confusion issues in traditional methods.

fusion network, and a **CMIM** module. First, we treat paired multimodal data as sequential temporal frames input to the model, where the first frame serves as the key frame and the remaining frames serve as auxiliary frames. The specific pipeline is as follows: First, the SAM2 image encoder is efficiently fine-tuned using LoRA+ to adapt to multimodal medical image features Ravi et al. (2024). Subsequently, all frames are sequentially fed into the progressive cross-modal fusion network: the key frame $F_{key} = \{X_i\}_{i=1}^{K}$ is first encoded into baseline features $f_{base}$, which are input to **MultiScaleNet** to initially generate target modalities. Then, each auxiliary feature $f_{aux_i}$ corresponding to auxiliary modality $X_i$ ($i > 1$) is sequentially processed through PreWeightNet, ThresholdNet, and EffiWeightNet to predict dedicated effective feature fusion weights for each source modality–target task pair $(i, j)$. This achieves fine-grained control over modality fusion through three hierarchical levels: global importance assessment, adaptive threshold gating, and local spatial modulation, thereby enabling progressive and effective information fusion. Finally, all $N$ target modalities $\{Y_j\}_{j=1}^{N}$ are generated through TaskHeadNet. Additionally, we introduce CMIM to ensure semantic consistency of modality identity in generated images through causal consistency constraints. The entire processing pipeline can be formulated as:

$$\mathcal{F} : \{X_1, X_2, \ldots, X_K\} \rightarrow \{Y_1, Y_2, \ldots, Y_N\} \tag{1}$$

where the input modality sequence is mapped to a unified feature space $\{F_i\}_{i=1}^{K} \in \mathbb{R}^{H \times W \times D}$ through the encoder.

## 2.1 MULTISCALENET

We conceptualize the $K$ heterogeneous medical modalities as temporal frames of the same anatomical structure under different imaging physics. Features extracted by the LoRA+-fine-tuned SAM2 encoder are processed through **MultiScaleNet**(fig 7), which handles the multi-scale feature outputs from SAM2. These features are first organized into a bottom-up feature pyramid structure, then processed through bidirectional Mamba modules employing a Fermat spiral scanning strategy for efficient and context-aware feature extraction Ma et al. (2024); Xing et al. (2024). The resulting features $F_{key}$ and $F_{aux_i}$ are subsequently passed to downstream processing modules.

The Fermat spiral scanning mechanism generates approximately isotropic attention distributions, enabling direction-unbiased spatial coverage while maintaining progressive continuity from center to periphery Yuan et al. (2025). The bidirectional state space model effectively eliminates inherent directional bias in medical image sequence modeling Heidari et al. (2024); Liu et al. (2024); Ma et al. (2024); Xing et al. (2024). This mitigates issues such as blurred lesion boundary identification and asymmetric cross-regional feature associations

caused by unidirectional modeling. The processed key frame and auxiliary frame features are then fed into their respective task head modules for subsequent generation tasks.

## 2.2 PreWeightNet for Task-Conditioned Weight Prediction

**PreWeightNet** is a key module in the Med-K2N framework responsible for predicting personalized global importance weights for each source modality–target task pair $(i, j)$. By learning the contribution of different modalities across various generation tasks, this module determines whether to activate specific modalities for fusion, providing reliable global priors for subsequent fusion and gating mechanisms while avoiding the one-size-fits-all weight allocation problem in multimodal fusion Havaei et al. (2016); Varsavsky et al. (2018). **PreWeightNet** employs a multi-reference information fusion architecture with a dedicated TaskAware Feature Encoder module that integrates and encodes multi-source input information, including baseline features (BaseFeature), auxiliary features (AuxFeature), and previous output-related information (PrevOutputs-related), to generate task-aware feature vectors $x_{ij}^{\text{task-aware}}$. This encoding process incorporates external memory modules to achieve task-oriented interaction and fusion of multi-source information Liao et al. (2025).

Furthermore, the system maintains a learnable parameter matrix $M_j \in \mathbb{R}^{D \times K}$ for each target task $j$, serving as a task-specific memory bank for storing successfully fused feature patterns, inspired by attention mechanisms in Transformer architectures Chen et al. (2021). Relevant memory items are retrieved through an attention mechanism:

$$m_j^{\text{retrieved}} = \sum_{k=1}^{K} \text{Softmax}\left(\frac{q_j \cdot M_j[:, k]}{\sqrt{D}}\right) \cdot M_j[:, k] \tag{2}$$

where $q_j = \text{TaskEncoder}(F_{\text{base}}, e_j^{\text{task}}, Q_{\text{context}})$ represents the task query vector constructed based on baseline features, task embeddings, and contextual information. Finally, the global importance weight $w_{ij}^{\text{global}}$ is obtained by fusing current task-aware features $x_{ij}^{\text{task-aware}}$ with retrieved historical experience memory $m_j^{\text{retrieved}}$, and output through an MLP with activation function:

$$w_{ij}^{\text{global}} = \sigma\left(\text{MLP}\left([x_{ij}^{\text{task-aware}}, m_j^{\text{retrieved}}]\right)\right) \tag{3}$$

## 2.3 ThresholdNet for Adaptive Threshold Learning

**ThresholdNet** extends the global contribution assessment from PreWeightNet by learning adaptive filtering thresholds that determine acceptance criteria for auxiliary modality information. As an intelligent decision-making component in the Med-K2N framework, this module learns a personalized acceptance threshold $\tau_{ij}$ for each source modality–target task pair $(i, j)$. By perceiving task-specific characteristics and inter-modal differences, it achieves selective filtering of beneficial information while suppressing low-value inputs Li et al. (2024a). Specifically, **ThresholdNet** constructs a dynamic adaptive gating mechanism by integrating task-related difficulty patterns and historical performance feedback, enabling a paradigm shift from "indiscriminate fusion" to "precision filtering."

In terms of architectural design, this module employs a lightweight network that integrates multi-source reference information through its Quality Gate Controller, including global weights $w_{ij}^{\text{global}}$ from PreWeightNet, task memory retrieval results $m_j^{\text{retrieved}}$, modality compatibility $C_{ij}$, and performance history $p_{ij}$ Ma et al. (2024). The controller performs feature extraction and collaborative fusion on this information, outputting fused feature vectors $x_{ij}^{\text{gate}}$ for threshold prediction. The adaptive threshold $\tau_{ij}$ is ultimately obtained through the following process:

$$x_{ij}^{\text{gate}} = \text{GateController}\left([w_{ij}^{\text{global}}, m_j^{\text{retrieved}}, C_{ij}, p_{ij}]\right) \tag{4}$$

$$\tau_{ij} = \tau_{\min} + (\tau_{\max} - \tau_{\min}) \times \sigma\left(\text{MLP}(x_{ij}^{\text{gate}})\right) \tag{5}$$

where $C_{ij} = \text{CompatEncoder}(e_i^{\text{modal}}, e_j^{\text{task}})$ represents modality compatibility encoding, and $p_{ij}$ denotes the historical performance statistics vector for the modality-task pair. The threshold bounds are set as $\tau_{\min} = 0.05$ and $\tau_{\max} = 0.9$.

## 2.4 EffiWeightNet for Efficient Weight Computation

**EffiWeightNet** learns and outputs final effective weight maps $w_{ij}^{\text{eff}}$ for feature fusion, building upon the global weights and adaptive thresholds provided by **PreWeightNet** and **ThresholdNet**. The core objective of this module is to compress multi-source information into reliable and continuous fusion control signals through a lightweight, end-to-end learnable network structure, thereby achieving a paradigm shift from traditional "rule-driven" to "data-driven" weight integration approaches Goodfellow et al. (2016). Specifically, **EffiWeightNet** introduces a learnable Fusion Weight Calculator module that comprehensively integrates all outputs from preceding modules and other relevant contextual information, including global weights $w_{ij}^{\text{global}}$ from PreWeightNet, adaptive thresholds $\tau_{ij}$, task memory retrieval results $m_j^{\text{retrieved}}$, gating features $x_{ij}^{\text{gate}}$, as well as task and modality embedding representations $c_j^{\text{task}}$ and $c_i^{\text{modal}}$ Liu et al. (2023). The calculator fuses multi-source inputs through linear or lightweight projection (Proj):

$$f_{\text{fused}} = \text{Proj}\left([w_{ij}^{\text{global}}, \tau_{ij}, m_j^{\text{retrieved}}, x_{ij}^{\text{gate}}, c_j^{\text{task}}, c_i^{\text{modal}}]\right) \tag{6}$$

Finally, the effective fusion weights are generated through a multi-layer perceptron (MLP) and Sigmoid activation function, with value ranges constrained using a clamp function to ensure numerical stability:

$$w_{ij}^{\text{eff}} = \text{clamp}\left(\sigma\left(\text{MLP}(f_{\text{fused}})\right), \epsilon, 1 - \epsilon\right) \tag{7}$$

where $\epsilon$ is a small lower bound (e.g., 0.001) used to prevent weights from reaching extreme values that could affect training stability Hinton et al. (2006). This design avoids the binary decision limitations of traditional hard threshold-based methods such as $w_{\text{eff}} = w_{\text{global}} \times I(\tau > \text{threshold})$, achieving more refined and adaptive fusion weight allocation.

## 2.5 TaskHeadNet

**TaskHeadNet**(fig 2) serves as the core module within the Med-K2N framework, responsible for final generation and quality control functions. This module adopts an innovative architecture of **concurrent multi-head generation–quality-driven selection–dynamic feedback**, integrating three major functionalities: feature fusion, multi-candidate generation, and quality feedback optimization Ravi et al. (2024). It achieves end-to-end mapping from weighted features to target modality images while establishing a complete quality-driven closed-loop optimization mechanism Chartsias et al. (2018). **TaskHeadNet** first performs unified encoding of multi-source input information through a shared feature fusion layer, including baseline features of key frames $f_{\text{base}}$, auxiliary features weighted by effective weights $w_{ij}^{\text{eff}} \odot f_i^{\text{en}}$, and task context embeddings $c_j^{\text{task}}$, mapping them to a unified generative representation space:

$$f_{\text{shared}} = \text{SharedEnc}\left(f_{\text{base}}, \sum_{i=2}^{K} w_{ij}^{\text{eff}} \odot F_i^{\text{en}}, c_j^{\text{task}}\right) \tag{8}$$

Based on this foundation, the module deploys $K_{\text{head}}$ independent and structurally diverse generation heads $\{\text{Head}_j^{(k)}\}_{k=1}^{K_{\text{head}}}$ for each target task $j$, generating multiple candidate images in concurrent and significantly enhancing the robustness of generation results through structural diversity and quality competition mechanisms:

$$\{Y_j^{(k)}\}_{k=1}^{K_{\text{head}}} = \{\text{Head}_j^{(k)}\left(\text{ModalAdapt}_j^{(k)}(f_{\text{shared}})\right)\}_{k=1}^{K_{\text{head}}} \tag{9}$$

The generated multiple candidate images are further subjected to quality-driven selection through an integrated quality assessment module. This evaluator comprehensively assesses candidate results from multiple dimensions including image clarity, modal consistency, anatomical structural integrity, and pathological feature preservation Staartjes et al. (2021), automatically selecting the candidate with the highest quality score as the final output:

$$Q_j^{(k)} = \text{QualityFeedback}\left(Y_j^{(k)}, X_{\text{ref}}, \text{ModalitySpec}_j\right), \quad Y_j^{\text{final}} = Y_j^{(\arg\max_k Q_j^{(k)})} \tag{10}$$

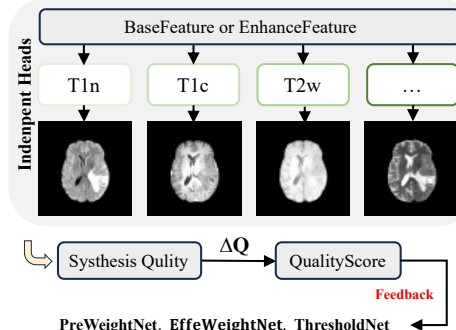

Figure 2: Architecture of TaskHeadNet, illustrating the concurrent multi-head generation, quality-driven selection, and dynamic feedback mechanisms.

**TaskHeadNet** establishes a quality feedback closed-loop optimization mechanism. Quality information $\Delta Q_j = Q_j^{\text{current}} - Q_j^{\text{previous}}$ is fed back to preceding modules: quality gains guide **PreWeightNet** updates, confidence distributions assist **ThresholdNet** threshold adjustment, and local error maps guide **EffiWeightNet** spatial weight optimization. This dynamic feedback forms a self-optimizing system that enhances adaptability and consistency in cross-modal multi-task medical image generation.

### 2.6 Causal Modality Identity Module (CMIM)

To address modal confusion in multi-modal medical image generation (e.g., T2-weighted signals appearing in generated T1-MRI, or synthetic CT images incorrectly retaining MRI contrast characteristics), we introduce the CMIM(fig 3) to enhance modal consistency and clinical safety of generated results. CMIM is built upon an explicit causal chain of **"modality type → image features → semantic expression"**. By modeling this causal relationship, it ensures generated images strictly follow the imaging characteristics and semantic constraints of the target modality, thereby suppressing modal feature confusion and reducing clinical risks from generation errors Rudie et al. (2019); Li et al. (2023b).

The module adopts a vision-text dual-encoder structure that maps generated images and corresponding modal text descriptions to the same semantic space Kirillov et al. (2023):

$$\mathbf{v_j} = \text{VisionEncoder}(Y_j), \quad \mathbf{t_j} = \text{TextEncoder}(D_j) \tag{11}$$

and constrains semantic consistency through cross-modal contrastive loss:

$$\mathcal{L}_{\text{cua}} = -\log \frac{\exp(\text{sim}(\mathbf{v_j}, \mathbf{t_j})/\tau)}{\sum_{k=1}^{N} \exp(\text{sim}(\mathbf{v_j}, \mathbf{t_k})/\tau)} \tag{12}$$

To further strengthen modal discriminative features, CMIM proposes a metric learning strategy based on causal inference. It uses generated images as anchors, real target modal images as positive samples, and other modal images as negative samples, enabling the model to focus on modal semantic essence rather than superficial differences. The metric loss function is defined as:

$$L_{metric} = \sum_{j=1}^{N} \max(0, \alpha + d(v_j^{gen}, v_j^{ref}) - d(v_j^{gen}, v_k^{neg})) \tag{13}$$

where $d(\cdot, \cdot)$ denotes feature distance and $\alpha$ is the margin parameter, ensuring generated images are close to correct modal references and distant from incorrect modal samples in feature space.

Med-K2N employs a multi-objective loss function that combines four complementary loss terms through weighted summation to balance generation performance across different levels:

$$L_{total} = \lambda_1 L_{L1} + \lambda_2 L_{SSIM} + \lambda_3 L_{causal} + \lambda_4 L_{metric} \tag{14}$$

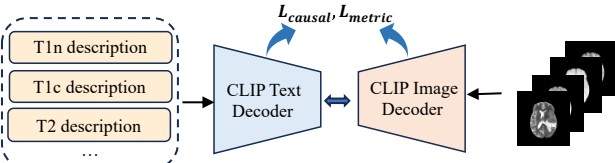

Figure 3: Architecture of the Causal Modality Identity Module (CMIM). This module establishes causal consistency constraints between modality description texts and generated images through CLIP dual encoders, utilizing contrastive loss and metric learning to prevent modality identity confusion.

## 3 EXPERIMENTS

### 3.1 DATASETS

**Combined Brain Tumor Dataset (BraTS2019 + BraTS-MEN + BraTS-MET)**: This dataset integrates three complementary datasets: BraTS2019, BraTS-MEN, and BraTS-MET, totaling 2,547 patients, including gliomas (795 cases), meningiomas (1,424 cases), and metastases (328 cases). Each case contains four MRI modality sequences: T1-weighted images (T1n), T1-contrast enhanced images (T1c), T2-weighted images (T2w), and FLAIR sequences (T2f).

**ISLES 2022 Dataset**: The ISLES 2022 dataset contains 400 expert-annotated multi-center MRI cases. Each case contains three MRI modality sequences: diffusion-weighted imaging (DWI, b=1000), apparent diffusion coefficient maps (ADC), and fluid-attenuated inversion recovery sequences (FLAIR).

### 3.2 RESULTS OF THE PROPOSED METHOD

In Figures 4, we present exemplary synthetic images produced by our method on the combined brain tumor dataset . The four-digit codes indicate the availability of the T1n, T1c, T2w, and T2f modalities, where '1' denotes available and '0' denotes missing. The results demonstrate that our progressive fusion strategy enables higher-quality image synthesis when more source modalities are available. For example, in synthesizing T2w images of brain tumors (third row of fig 4), using T1n alone yields blurry tumor boundaries and poor contrast enhancement. Integrating additional modalities significantly improves the visual fidelity and structural accuracy of the synthesized images.

Tables 1 and 2 provide quantitative results under various input-output settings on the two datasets. Both tables confirm that leveraging all available modalities to synthesize the target modality achieves the best performance in terms of PSNR and SSIM, consistent with the qualitative observations Chartsias et al. (2017); Dar et al. (2019); Liu et al. (2023).

### 3.3 ABLATION STUDY

Ablation studies are conducted to systematically evaluate the effectiveness of key modules in the Med-K2N model and investigate the optimal hyperparameter configuration for the progressive fusion mechanism. All experiments are performed on a merged brain tumor dataset, with the task defined as generating T2f modality images from T1n, T1c, and T2w inputs.

Effectiveness of all modules: Table 3 assesses the contribution of each core module in Med-K2N, integrating components sequentially in the order of "baseline model → weight prediction → threshold filtering → effective weight reconstruction → cross-modal interaction → curriculum learning" to construct a comprehensive causal contribution chain analysis. Experimental results demonstrate that **PreWeightNet**(B1), through its adaptive weight allocation mechanism, improves PSNR by 0.52 dB compared to simple average fusion, validating the importance of modality-specific weighting Havaei et al. (2016); Varsavsky et al. (2018).

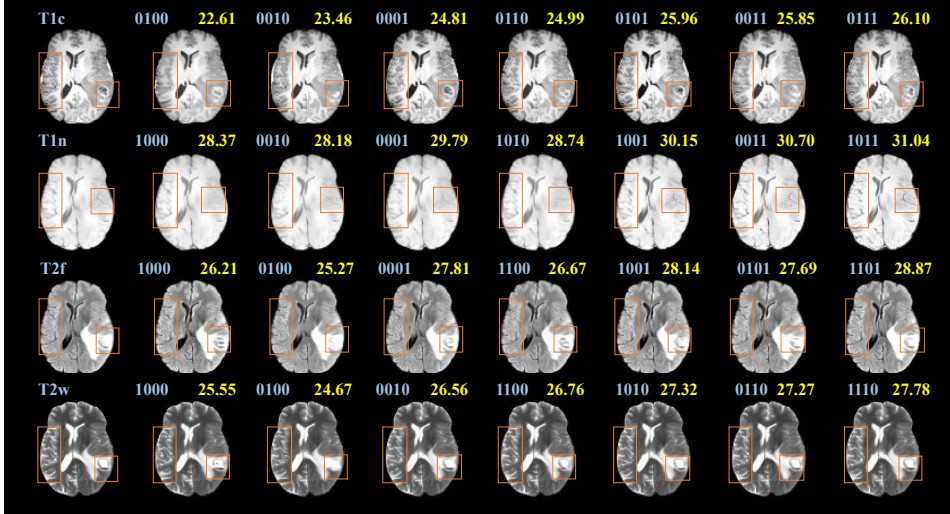

Figure 4: Synthesis results on the Combined Brain Tumor Dataset. The figure demonstrates generation performance across four target modalities: T1c, T1n, T2f, and T2w.

Table 1: Quantitative comparison results of our method and other unified synthesis methods on the Combined Brain Tumor dataset. The results with * indicate $p < 0.05$ compared with our method based on Wilcoxon signed-rank test. Results are ordered by generation difficulty.

| Available modalities | | | | Results (PSNR↑, SSIM↑) | | | |
|---|---|---|---|---|---|---|---|
| T1n | T1c | T2w | T2f | MM-Synthesis Chartsias et al. (2017) | pGAN Dar et al. (2019) | MM-Transformer Liu et al. (2023) | Med-K2N(ours) |
| | ✓ | | | 22.18*, 0.854* | 22.85*, 0.862* | 23.72*, 0.875* | **24.33, 0.883** |
| | | ✓ | | 25.22*, 0.891* | 25.84*, 0.898* | 26.45*, 0.906* | **27.05, 0.912** |
| | | | ✓ | 24.85*, 0.882* | 25.12*, 0.886* | 25.68*, 0.895* | **26.21, 0.901** |
| | ✓ | ✓ | | 23.89*, 0.872* | 24.51*, 0.881* | 25.28*, 0.892* | **25.85, 0.898** |
| | ✓ | | ✓ | 23.45*, 0.868* | 24.12*, 0.876* | 24.98*, 0.887* | **25.61, 0.894** |
| | | ✓ | ✓ | 25.78*, 0.905* | 26.34*, 0.912* | 27.12*, 0.923* | **27.68, 0.929** |
| | ✓ | ✓ | ✓ | 24.68*, 0.885* | 25.42*, 0.893* | 26.35*, 0.904* | **26.98, 0.910** |
| ✓ | | | | 27.65*, 0.925* | 28.21*, 0.932* | 28.89*, 0.941* | **29.46, 0.947** |
| ✓ | ✓ | | | 25.84*, 0.903* | 26.48*, 0.911* | 27.35*, 0.922* | **27.92, 0.927** |
| ✓ | | ✓ | | 27.89*, 0.928* | 28.45*, 0.935* | 29.21*, 0.944* | **29.78, 0.949** |
| ✓ | | | ✓ | 28.12*, 0.931* | 28.78*, 0.938* | 29.58*, 0.947* | **30.15, 0.952** |
| ✓ | | ✓ | ✓ | 28.45*, 0.934* | 29.12*, 0.941* | 29.95*, 0.950* | **30.58, 0.955** |
| ✓ | ✓ | | ✓ | 26.58*, 0.915* | 27.24*, 0.922* | 28.15*, 0.932* | **28.73, 0.937** |
| ✓ | ✓ | ✓ | | 26.21*, 0.911* | 26.89*, 0.918* | 27.82*, 0.928* | **28.41, 0.933** |

**ThresholdNet**(B2) introduces suppression of low-contribution regions on top of B1, yielding a marginal gain of 0.16 dB, indicating that filtering redundant modal information enhances reconstruction quality. **EffiWeightNet**(B3) further achieves a performance improvement of

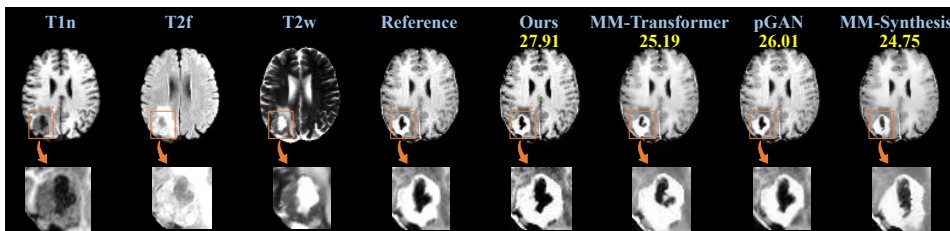

Figure 5: Representative qualitative visual comparisons of T1c images synthesized from T1n, T2w, and T2f sequences on the Combined Brain Tumor Dataset. Orange boxes highlight key reconstructed regions, with corresponding zoomed-in views provided. Yellow values indicate PSNR scores.

Table 2: Quantitative comparison results of our method and other unified synthesis methods on the ISLES 2022 dataset. The results with * indicate $p < 0.05$ compared with our method based on Wilcoxon signed-rank test.

| Available modalities | | | Results (PSNR↑, SSIM↑) | | | |
|---|---|---|---|---|---|---|
| ADC | DWI | FLAIR | MM-Synthesis | pGAN | MM-Transformer | Med-K2N(Ours) |
| | ✓ | | 22.15*, 0.845* | 22.78*, 0.852* | 23.42*, 0.861* | **24.55**, **0.878** |
| ✓ | | | 24.89*, 0.901* | 25.38*, 0.908* | 25.62*, 0.915* | **26.72**, **0.931** |
| | | ✓ | 24.12*, 0.885* | 24.65*, 0.892* | 25.01*, 0.899* | **26.12**, **0.915** |
| ✓ | ✓ | | 23.45*, 0.862* | 23.89*, 0.869* | 24.17*, 0.876* | **24.32**, **0.882** |
| ✓ | | ✓ | 25.95*, 0.918* | 26.42*, 0.925* | 26.54*, 0.932* | **27.65**, **0.948** |
| | ✓ | ✓ | 22.78*, 0.832* | 23.19*, 0.839* | 23.56*, 0.846* | **24.89**, **0.861** |
| ✓ | ✓ | ✓ | 24.52*, 0.878* | 24.98*, 0.885* | 25.21*, 0.892* | **26.21**, **0.908** |
| | | ✓ | 25.28*, 0.912* | 25.78*, 0.919* | 26.21*, 0.926* | **27.79**, **0.942** |
| | ✓ | ✓ | 24.89*, 0.896* | 25.32*, 0.903* | 25.76*, 0.910* | **26.32**, **0.926** |

0.68 dB by optimizing weight distribution through spatial context fusion, highlighting the effectiveness of local-global feature synergy. **CMIM**(B4) achieves a notable gain of 0.39 dB without introducing additional parameter costs, demonstrating the value of interaction modeling in multimodal fusion Chen et al. (2021). **The curriculum learning strategy** (B5) significantly improves convergence stability through optimized training scheduling, resulting in a final performance gain of 0.13 dB without extra inference overhead.

## 4 DISCUSSION

The proposed Med-K2N framework addresses key challenges in medical cross-modal synthesis Rudie et al. (2019); Li et al. (2023b). Our progressive fusion strategy and quality-driven selection mechanism prevent information dilution Jog et al. (2015); Roy et al. (2013). Consistent PSNR improvements across different K→N configurations validate this approach compared to traditional fusion methods Chartsias et al. (2018); Sharma & Hamarneh (2019). The CMIM module prevents modality identity confusion through causal consistency constraints. This contribution addresses clinical safety concerns Li et al. (2023b); Rudie et al. (2019). Modality confusion in synthetic images may cause diagnostic errors, which is particularly critical in medical AI applications Rudie et al. (2019); Staartjes et al. (2021). Several limitations exist in this work. Computational complexity increases with input modality number. This may limit real-time clinical applications. Our evaluation focuses on brain imaging datasets. Broader validation across anatomical regions and pathological conditions is required Pan et al. (2018); Bowles & et al. (2016); Iglesias et al. (2013); Jog et al. (2015). Future research directions include lightweight architecture development and uncertainty quantification methods Rudie et al. (2019); Li et al. (2023b). The K→N flexibility addresses clinical scenarios with variable modality availability. Clinical validation studies remain necessary to establish diagnostic equivalence between synthetic and acquired images Li et al. (2023b); Staartjes et al. (2021).

Table 3: Ablation study results on merged brain tumor dataset for T2f modality generation

| Stage | Configuration | PSNR(SSIM) ↑ | Stage Gain |
|---|---|---|---|
| B0 | Baseline Fusion | 26.53(0.878) | - |
| B1 | +Weight Prediction | 27.05(0.895) | +0.52 |
| B2 | +Threshold Filtering | 27.21(0.902) | +0.16 |
| B3 | +Effective Weight | 27.89(0.919) | +0.68 |
| B4 | +CMIM Interaction | 28.28(0.929) | +0.39 |
| B5 | +Curriculum Learning | 28.41(0.933) | +0.13 |

## 5 ETHICS STATEMENT

Our contributions enable advances in medical cross-modal image synthesis, with potential to significantly improve clinical diagnostic workflows by reconstructing missing imaging modalities from available data. This technology could benefit healthcare by reducing patient scan times, radiation exposure, and providing diagnostic support in resource-limited settings. While we do not anticipate specific negative impacts from this work, synthetic medical images require careful clinical validation and regulatory approval before deployment. As with any medical AI tool, there is potential for misuse if applied without proper validation or beyond validated scope. We strongly emphasize that our framework is intended as a research tool and clinical decision support, not as replacement for standard imaging protocols, and encourage the medical community to prioritize patient safety and maintain transparency about synthetic image usage when applying these technologies in clinical practice.

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

# A    APPENDIX

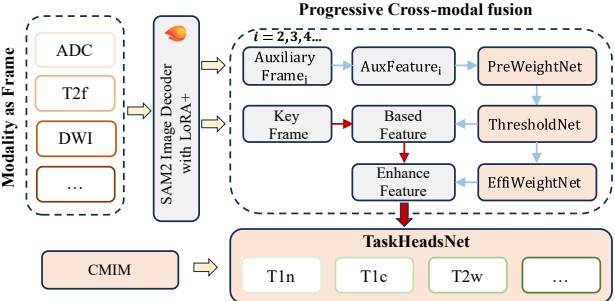

Figure 6:   Overall of Med-K2N

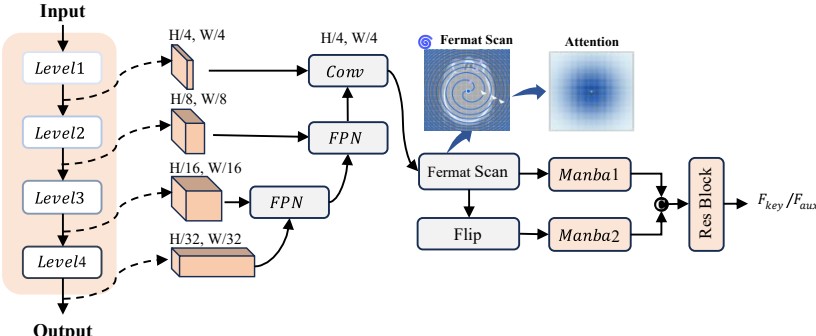

Figure 7: MultiScaleNet architecture for processing multi-scale features from the SAM2 encoder. It employs bidirectional Mamba modules with Fermat spiral scanning for efficient feature extraction.

## A.1    MORE EXPERIMENTAL DETAILS

**Implementation details:** We implement our model using PyTorch and train on NVIDIA A100 GPUs. The student SAM2 encoder is initialized from a checkpoint provided by the authors Ravi et al. (2024), while other modules are randomly initialized using Kaiming initialization **?**. During preprocessing we apply the same augmentation pipeline used by our training scripts: horizontal flipping with 0.5 probability, gentle color jitter, Gaussian blurring (p = 0.1), and random resized cropping with scale between 0.8 and 1.2 before identity normalization. All MedicalMRI slices are resampled to $256 \times 256$, aligning the student SAM2 encoder's receptive field. To stay within a single 24 GB GPU, we keep the LoRA rank at $r = 16$ and train with a per-GPU batch of 48 while accumulating gradients for three steps, which mirrors a larger effective batch without exhausting memory. Training runs for 100 epochs under a cosine learning-rate schedule starting at $1 \times 10^{-4}$; when expanding to tasks with more target modalities we slightly increase the initial learning rate (e.g., $+10\%$) to avoid underfitting.

**curriculum study:** We further employ a curriculum strategy that mirrors the k→n task difficulty encoded in 'train.py': the 100 training epochs are split using ratios $(0.2, 0.2, 0.3, 0.3)$ into four stages—easy, medium, hard, and expert. Early epochs (easy) sample only cross-modal $1 \to 1$ mappings while explicitly excluding identity targets; the medium phase introduces multimodal fusion ($k \to 1$) to reinforce aggregation. During the hard stage the sampler flips to $1 \to k$ expansions so the generator learns to hallucinate missing contrasts, and the final expert stage activates full $k \to t$ patterns with strict non-overlapping input/target sets. The stage controller is reproducible (seeded per epoch/batch/rank) and

can optionally extend to validation, though we disable that by default to preserve unbiased metrics. YAML overrides in the 'CURRICULUM' block (or the '–curriculum-ratios' flag) let us retune phase lengths, while the loss mask linked to each stage progressively enables perceptual, causal, and quality-aware terms alongside the base SSIM/L1 objectives.

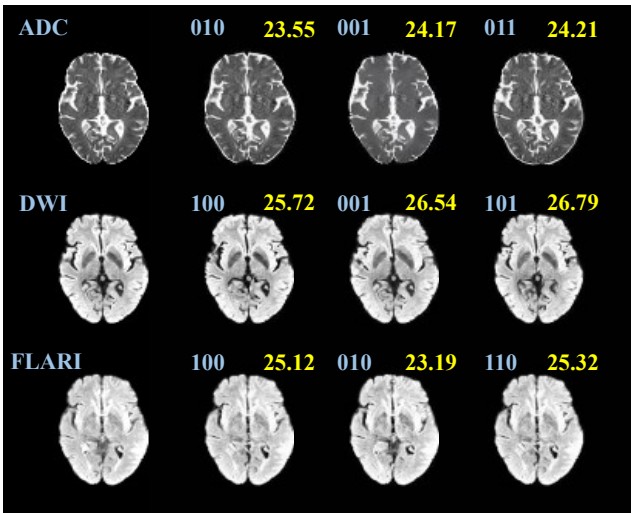

Figure 8: Synthesis results on the ISLES 2022 Dataset. The figure illustrates the generation performance for three target modalities: ADC, DWI, and FLAIR. The three-digit binary code above each image indicates the availability status of the three input modalities ("1" denotes available, "0" denotes missing). Yellow numbers display the corresponding PSNR values. The results demonstrate that PSNR values exhibit an ascending trend as the number of available input modalities increases, validating the effectiveness of multi-modal fusion.

