# OpenReview forum: "Med-K2N: Flexible K-to-N Modality Translation for Medical Image Synthesis"
_ICLR.cc/2026/Conference — ICLR 2026 Conference Withdrawn Submission_

### Official Review · Reviewer_umez · 2025-10-19

**Soundness:** 3
**Presentation:** 2
**Contribution:** 2
**Rating:** 2
**Confidence:** 5

**Summary:**

This work proposed Med-K2N, a unified framework for multi-modal translation. Based on a LoRA-adapted SAM2 encoder, Med-K2N incorporates three collaborative fusion modules and an identity module to ensure semantic consistency across generated modalities. The framework takes multi-modal inputs as sequential frames and progressively fuses them through a quality-aware enhancement strategy, adaptively learning modality-specific contributions and filtering redundant information. Evaluations on the two datasets demonstrate that Med-K2N consistently outperforms state-of-the-art methods in PSNR and SSIM.

**Strengths:**

Unified multi-modal translation is important for clinical applications.

Given its flexible uses, adaptive learning among different modalities is reasonable

Better performance than several existing works.

**Weaknesses:**

The paper introduces many modules, each contributing only marginal performance improvements according to the ablation study. Although the overall framework is adaptive, it remains difficult to grasp the key insights and the underlying technical rationale. A strong technical paper typically focuses on one or two main contributions, while other components should support the central idea rather than appear as loosely connected extensions.

After reviewing the anonymized code repository, several inconsistencies and missing details raise concerns about the reproducibility and validity of the contributions. For instance, the implementation of the L1 loss focuses on high-intensity regions, which is not discussed in the manuscript. The implementation section also lacks essential information, such as the motivation for merging different brain tumor datasets, the criteria for dataset splitting, and the procedures used to ensure fair and consistent preprocessing.

The related work section is not sufficient about the existing literature on unified medical modality translation. Several prior works, including [1], have addressed similar challenges of modeling both modality-specific and modality-shared representations in a unified framework. The current comparisons are limited, and the chosen metrics alone cannot convincingly demonstrate the superiority of the proposed method. Incorporating downstream task evaluations is essential and can provide stronger evidence of practical benefits.
[1] Unified Multi-Modal Image Synthesis for Missing Modality Imputation, IEEE Transactions on Medical Imaging, 2024.

While SAM2 is a powerful vision foundation model, its usage for medical image translation is questionable. SAM2 is primarily designed for perceptual understanding, while medical translation requires complete recovery of fine-grained structural and intensity information. In addition, the significant domain gap between natural RGB images and medical grayscale data should be discussed further. The paper can be improved by a clearer analysis of these differences and a key justification of how SAM2 can be effectively adapted for this task.

**Questions:**

See the above weaknesses. Overall, the paper requires substantial revision.

---

### Official Review · Reviewer_qaQi · 2025-10-31

**Soundness:** 3
**Presentation:** 3
**Contribution:** 2
**Rating:** 4
**Confidence:** 4

**Summary:**

This paper proposes Med-K2N, a quality-aware progressive fusion framework for medical image synthesis.

**Strengths:**

1) Drawing inspiration from SAM2’s sequential frame paradigm, the model treats K modal images as sequential frames, inputs them one by one, and then sequentially optimizes the results of N target modalities.

2) To address the three core challenges proposed in the study, the author designs three collaborative modules (PreWeightNet, ThresholdNet, and EffiWeightNet), each tailored to solve one of these challenges, respectively.

3) The experimental results show the effectiveness of the proposed model.

**Weaknesses:**

1) The experiment has limitations: Only three comparative methods are included; The ablation experiments are only conducted on the Combined Brain Tumor Dataset, and the experimental results are only presented for the T2f modality generation task.

2) Unclear descriptions: At the end of Section 2 (Method), it is mentioned that CMIM maps data to F. Is this role the same as that of V in Section 2.6? The source of q in the calculation of m_j^retrieved in Section 2.2 is unclear, specifically regarding the task embedding and Q_context. This issue also appears in Sections 2.3 and 2.4; in addition, the symbol setting of embeddings is confusing, are e_j^task and c_j^task the same?

3) Is the experimental result related to the selection of the key frame? Is the input order related to the output quality? Since the idea of SAM2’s sequential input is adopted, this issue should be discussed.

4) The function of CMIM is to align the descriptions of the same modality with image features. However, it is unclear what the specific modality descriptions are. If they are merely simple terms such as "T1c, T2f, etc.", will the output results of the CLIP text encoder for these terms with little difference be very similar?

**Questions:**

Please see "Weakness" section.

---

### Official Review · Reviewer_ukKy · 2025-11-01

**Soundness:** 3
**Presentation:** 3
**Contribution:** 2
**Rating:** 4
**Confidence:** 4

**Summary:**

The paper introduces Med-K2N, a flexible K-to-N cross-modal medical image synthesis framework that treats multiple input modalities as “sequential frames” and progressively fuses them to generate multiple target modalities. The system is built around three collaborative modules:

•	PreWeightNet: estimates global modality-to-task contribution weights and retrieves task-specific fusion patterns via a memory bank.

•	ThresholdNet: learns adaptive acceptance thresholds to gate auxiliary modality information based on task difficulty and historical performance.

•	EffiWeightNet: produces final effective spatial weight maps for fusion, integrating global weights, thresholds, memory retrieval, gating features, and task/modality embeddings.

A TaskHeadNet performs concurrent multi-head generation and quality-driven selection with closed-loop feedback to upstream modules. To enforce modality identity consistency, the Causal Modality Identity Module (CMIM) employs a vision-text dual-encoder (CLIP-like) with contrastive and metric losses to align generated images with target modality descriptions and discourage “modality confusion.”
The backbone is a LoRA+-fine-tuned SAM2 encoder; multi-scale features are processed by a MultiScaleNet using bidirectional Mamba modules and a Fermat spiral scanning strategy. Experiments on a combined brain tumor dataset (BraTS variants) and ISLES 2022 demonstrate consistent PSNR/SSIM improvements over prior unified synthesis methods (MM-Synthesis, pGAN, MM-Transformer), with ablations attributing gains to each module in the progressive pipeline.

**Strengths:**

Originality:
•	The explicit decomposition of fusion into three stages (global contribution prediction, adaptive gating, and efficient local weighting) is a thoughtful response to clinical heterogeneity in K→N synthesis. Framing modality contributions at both coarse and fine levels and adding quality-driven selection is more nuanced than conventional uniform fusion.

•	Treating multimodal inputs as sequential frames and leveraging a SAM2 encoder with LoRA+ fine-tuning, paired with Mamba-based MultiScaleNet and Fermat spiral scanning, is a creative assembly of recent architectural ideas tailored to medical synthesis.

•	CMIM’s causal consistency constraint—grounding generated modality identity via vision-language alignment plus metric learning against real references—is a meaningful attempt to address a real clinical failure mode (modality confusion).

Quality:
•	The system design is comprehensive: encoder adaptation, multi-scale feature handling, progressive fusion decision-making, multi-head generation with quality evaluation, and feedback. The ablation table traces incremental gains, suggesting each component contributes.

•	The experiments cover multiple K→N configurations and report consistent PSNR/SSIM trends with more input modalities. Wilcoxon signed-rank tests are used to claim statistical significance relative to baselines.
•	Implementation details, curriculum scheduling, and training hyperparameters are provided in the appendix, which aids reproducibility.

Clarity:
•	The motivation is well articulated: not all modalities contribute equally to all targets; indiscriminate fusion dilutes signal; multi-output generation needs explicit identity control.

•	Module roles and data flow are explained in a way that a reader can follow the overall pipeline. The figures capture the progressive fusion idea and the quality gate concept reasonably well.


Significance:
•	Flexible K→N synthesis is practically relevant in clinical workflows where modality availability varies and multi-output reconstruction can reduce repeated scans or fill gaps. The approach could influence future multimodal synthesis work and may generalize to other anatomical regions and tasks.

**Weaknesses:**

Limited benchmarking scope and depth:

•	Comparisons are restricted to three prior unified synthesis baselines. The field contains many strong and recent methods for cross-modal synthesis and multimodal fusion (e.g., cycle-consistent GANs tailored to medical imaging, diffusion-based cross-modal generators, transformer-based cross-modal mappings beyond MM-Transformer, flow-guided or registration-aware synthesis). Without a broader set of baselines—including diffusion and modern transformer/diffusion hybrids—the strength of the claims is hard to calibrate.

•	The backbone choice (SAM2 + LoRA+) and MultiScaleNet (Mamba + Fermat spiral) likely contribute substantially to performance. The paper does not include ablations on the encoder choice (e.g., SAM2 vs. a standard medical encoder) nor on MultiScaleNet vs. simpler pyramids. It is difficult to isolate how much the gains come from backbone capacity versus the proposed fusion modules.


Methodological detail gaps and potential over-engineering:

•	PreWeightNet’s memory bank Mj and retrieval mechanism are described at a high level, but the formation, size, update policy, and regularization for memory are not specified concretely. How memory is seeded and prevented from collapsing or overfitting remains unclear.
•	ThresholdNet uses bounds τmin and τmax and fuses multiple signals (compatibility, performance history), but the definitions of compatibility Cij and performance statistics pij are underspecified. How these signals are computed and updated online (and whether they introduce leakage or bias) needs more detail.
•	EffiWeightNet fuses many inputs via lightweight projections; there is no analysis of stability when signals disagree (e.g., high global weight but low threshold) or of calibration of the final weight maps across spatial scales.
•	TaskHeadNet’s “structurally diverse heads” and the quality evaluator are key to final outputs, yet their architectures and quality metrics are not fully specified. The quality assessment appears composite (clarity, modality consistency, anatomical integrity, pathology preservation), but the exact implementation (learned vs. hand-crafted, supervision, validation) is not detailed. This is a critical piece because the closed-loop depends on reliable scoring.

CMIM assumptions and reliance on vision-language priors:

•	The modality identity enforcement relies on text descriptions and CLIP-like encoders. The choice and quality of modality-specific textual prompts/descriptions for MRI contrasts (T1n/T1c/T2w/FLAIR) and CT vs. MRI are nontrivial. If the text prompts are simplistic, the system may learn superficial correlations. The paper does not discuss how the text descriptions were constructed or validated by radiologists, nor the domain shift between CLIP’s natural image pretraining and medical imagery.

•	The metric loss uses generated images as anchors, real target modality images as positives, and “other modal images” as negatives. Without careful sampling and alignment, this can be confounded by anatomy/pathology variations rather than modality semantics.

Evaluation limitations and missing analyses:
•	Only brain MRI and ISLES are covered. No CT or multimodal MRI→CT synthesis experiments are shown (despite claiming CT-related heterogeneity in the intro). Generalization to other organs and modalities is acknowledged as future work but missing here.

•	The report focuses on PSNR/SSIM. Clinical fidelity often benefits from task-specific metrics (e.g., segmentation performance on synthetic images, radiologist-rated diagnostic utility, lesion conspicuity). There is no task-based downstream evaluation to demonstrate that the gains translate to diagnostic usefulness.

•	Computational cost vs. K and N isn’t quantified beyond high-level comments. The framework introduces multiple modules and multi-head generation. A clean complexity breakdown (params, FLOPs, latency per modality addition, memory footprint) and a comparison to baselines would be important, especially given claims of scalability.

Clarity and consistency issues:
•	Some figures and notations are cluttered or contain artifacts, and a few portions in the main figures look garbled (likely PDF extraction issues). While the narrative is understandable, the mathematical details are at times hand-wavy.

•	The “modality as frames” analogy is appropriate, but the sequential order and the dependence on ordering are not discussed. If the order of auxiliary modalities changes, do the weights differ? Is the system order-invariant?

**Questions:**

1.	Baselines and breadth:
•	Please include stronger and more diverse baselines: a recent diffusion-based medical cross-modal synthesis method; a cycle-consistent GAN tailored for MRI contrast synthesis; a transformer-based cross-modal mapping beyond MM-Transformer; and a simple strong baseline (e.g., per-modality U-Net with concatenated inputs and a learned fusion layer). This will better contextualize your gains.

2.	Encoder and MultiScaleNet ablations:
•	How much of the improvement stems from SAM2 + LoRA+ and the Mamba/Fermat spiral MultiScaleNet? Provide ablations replacing SAM2 with a standard medical encoder and MultiScaleNet with a simpler FPN. Also test SAM2 without LoRA+ fine-tuning to quantify adaptation benefits.

3.	Memory bank and ThresholdNet details:
•	For PreWeightNet, detail memory initialization, size per task, update rules, and any regularization to prevent drift or collapse. For ThresholdNet, define Cij (compatibility) and pij (performance statistics) precisely: how are they computed, updated, and normalized? Are they per-patient, per-task, or global?

4.	TaskHeadNet and quality evaluator:
•	Describe each generation head architecture and its diversity rationale. For the quality evaluator, specify metrics, supervision, and calibration. Is the evaluator trained jointly? How do you avoid circularity (e.g., evaluator favoring heads that mimic its preferred artifacts)? Please provide sensitivity analyses.

5.	CMIM prompts and domain alignment:
•	How are modality description texts constructed? Were they curated with radiologist input? Have you validated that the CLIP-like encoders provide reliable embeddings for medical contrasts? Consider reporting retrieval accuracy (matching images to modality descriptions) and ablations using more medical-specific vision-language models.

6.	Modality order and invariance:
•	Is the fusion process invariant to the order of auxiliary modalities? If not, justify the chosen ordering and analyze its effect. If yes, please provide details on how order invariance is achieved.

7.	Clinical utility and downstream tasks:

•	Beyond PSNR/SSIM, can you demonstrate that synthetic images improve downstream tasks (e.g., segmentation or detection) compared to baseline synthetic methods? Even a small study would strengthen clinical relevance.

8.	Complexity and scalability:

•	Provide a detailed breakdown of parameters, FLOPs, and latency for different K and N. How does the system scale if we go from K=2 to K=4 inputs? Are there bottlenecks? Any preliminary results on lightweight variants (smaller backbones, pruning, AMP/quantization)?

---

### Official Review · Reviewer_9HQq · 2025-11-07

**Soundness:** 3
**Presentation:** 3
**Contribution:** 3
**Rating:** 6
**Confidence:** 3

**Summary:**

This paper presents Med-K2N, a new medical image synthesis framework with the ability to generate flexible K-to-N image modalities. To tackle the inherent difficulties in parallel multi-output generation settings, it introduces explicit mechanisms for (1) modality-task-specific weighting for heterogeneity, (2) fusion quality control, and (3) enforcing correct modality identity consistency via pre-trained medical VLMs constraints. Comprehensive experiments on Brain Tumor and ISLES 2022 datasets shows state-of-the-art performances.

**Strengths:**

The study of flexible K-to-N modality synthesis is interesting and valuable. Previous works suffer from stringent requirement of fixed K→1 mappings, fail to account for modality difference, and prone to produce wrong modality identity. This work introduces principled, multi-level fusion strategy instead of naive concatenation, which brings clear generation improvement. Additionally, the causal VLM constraints is highly useful when tackling the modality identity issue. Overall, I find this work well-grounded, thoroughly designed, and show solid experiment results.

**Weaknesses:**

1. The design of this work seems quite heavy & complex. Beyond inserting LoRA into the based SAM2 module, it also introduces a MultiScaleNet, three weighting modules, multi-head generators, and a Causal Modality Identity Module. It is good to see that each component brings meaningful boost in Table 3, but is this pipeline scalable? What is the inference-time latency & memory footprint for real clinical deployment?

2. Right now only MRI image domain is studied (the Combined Brain Tumor Dataset as well as the ISLES 2022 Dataset only contain MRI data). To further enhance this work, study on other domain, like CT or X-Ray, is encouraged.

3. Right now only three baselines are used for comparison, and they are relatively old (dated back to 2017, 2019, and 2023). More recent works, like cross-conditioned diffusion[1] or structure-aware translation [2], should also be included, discussed, and compared against.

**References**

[1] Xing, Zhaohu, et al. "Cross-conditioned diffusion model for medical image to image translation." International Conference on Medical Image Computing and Computer-Assisted Intervention. Cham: Springer Nature Switzerland, 2024.

[2] Zhang, Xinzhe, et al. "Structure-Aware MRI Translation: Multi-modal Latent Diffusion Model with Arbitrary Missing Modalities." International Conference on Medical Image Computing and Computer-Assisted Intervention. Cham: Springer Nature Switzerland, 2025.

**Questions:**

1. The paper employs a TaskHeadNet for “quality assessment” to pick best candidate. I wonder how is this module designed and trained? I cannot find any related details. Is there any risk of overfitting the selector or leaking label information?

2. Does Med-K2N provide any indication when synthesized outputs are unreliable (e.g., hallucination, severe artifacts)?

3. In line 37-44, the authors claim "Cross-modal generation techniques are thus important for reconstructing missing modalities". I wonder, from the perspective of information theory, isn't the information presented in the generated images inherently upper bounded by the source input? How is this going to help clinical applications?

---

### Note · Authors · 2025-11-12

I have read and agree with the venue's withdrawal policy on behalf of myself and my co-authors.